

# Expression characterization of the guanylate-binding protein gene family in breast cancer and its association with the immune microenvironment

Min Wei[1], Peng Sun[2], Xuemei Liu[1], Xuhua Liu[1] and Jie Lei[3]

[1] Department of Science and Education, Nanshan Maternity and Child Healthcare Hospital, Shenzhen, China
[2] Administration Office, Nanshan Maternity and Child Healthcare Hospital, Shenzhen, China
[3] Laboratory Medicine, Nanshan Maternity and Child Healthcare Hospital, Shenzhen, China

Corresponding author
Min Wei, sznsfywm@163.com

## ABSTRACT

**Background**. Considering the complexity of prognostic assessment in breast cancer (BRCA) and the potential role of guanylate-binding protein (GBP) genes in immune regulation, the present research was designed to construct a prognostic model using GBP-related genes and to explore the mechanism of their role in BRCA.

**Methods**. Using RNA sequencing data of BRCA from public databases, GBP gene scores of BRCA samples were computed by single-sample GSEA (ssGSEA). The gene modules associated with the scores were identified by weighted gene co-expression network analysis (WCGNA). Subsequently, differentially expressed genes (DEGs) between BRCA samples and paraneoplastic samples were screened by the "limma" package and intersected with the modular genes. Key prognostic genes were further compressed by least absolute shrinkage and selection operator (LASSO), univariate and multivariate stepwise regression analyses were used to develop a risk model. Next, differences in the biological characteristics and immune infiltration between different risk groups were explored. Particularly, CCK-8, wound healing and transwell test were performed to examine the biological role of *DACT2* in BRCA.

**Results**. Low GBP scores in BRCA patients were significantly linked to a poorer overall survival. Two gene modules were closely and positively correlated with GBP scores, and their module genes were enriched in immune-related pathways. Subsequently, four key genes (*PSME2*, *DACT2*, *PIGR* and *STX11*) were screened to construct a risk model, which showed strong diagnostic performance. Notably, the infiltration of immune cells (*e.g.*, CD4/CD8 T cells, mast cells, and macrophages) was higher in low-risk BRCA patients. In addition, the RiskScore was significantly negatively correlated with ESTIMATE scores. *In vitro* cellular experiments demonstrated that *PSME2* was significantly upregulated in BRCA cell lines, while *DACT2*, *PIGR*, and *STX11* were all markedly downregulated in BRCA cells. In particular, overexpression of *DACT2* markedly suppressed the survival, migration, and invasion of AU565 and MDA-MB-231 cells.

**Conclusion**. This study constructed a prognostic model for BRCA based on GBP-related genes. The model was closely related to the immune microenvironment, contributing to the prognostic assessment and individualized treatment guidance in the management of BRCA.

## INTRODUCTION

Breast cancer (BRCA) accounts for more than 24% of new cancer cases worldwide and approximately 15% of tumor-related death cases (*Haikal et al., 2024*; *Xavier et al., 2023*). The standard treatment for BRCA is neoadjuvant chemotherapy, and radiotherapy is also a crucial part of its comprehensive treatment. Prospective randomized trials, meta-analyses, and observational data showed that postoperative radiotherapy can lower the recurrence and mortality rates of BRCA patients (*Whelan et al., 2015*; *Poortmans et al., 2015*; *Naeimzadeh et al., 2023*). However, the majority of BRCA patients remain incurable due to the self-renewal and differentiation of BRCA stem cells, which contribute to the recurrence, metastasis, and resistance to treatment of cancer (*Morel et al., 2008*; *Creighton et al., 2009*). The complexity of therapeutic responses and tumor heterogeneity further complicates accurate prognosis prediction for BRCA. Currently, prognostic evaluation for patients suffering from BRCA primarily relies on clinical and pathologic assessments based on TNM staging and molecular subtypes. To further reduce the mortality of BRCA, an accurate evaluation of patients' prognosis is required (*Duffy et al., 2020*).

Guanylate-binding proteins (GBPs) are a subfamily of interferon-γ (IFN-γ)-inducible GTPases that hydrolyze guanosine triphosphate to guanosine monophosphate and guanosine diphosphate (*Ghosh et al., 2006*). Humans contain seven well-characterized GBP proteins (GBP1-7) with molecular weights between 67 and 73 kDa (*Tretina et al., 2019*). GBPs, including GBP1 and GBP2, play crucial roles in innate immunity, host defense against pathogens, and antibacterial/antiviral properties during host anti-inflammatory and anti-infective defenses (*Vestal & Jeyaratnam, 2011*; *Honkala, Tailor & Malhotra, 2019*). A study highlighted the significance of immune microenvironment in tumor biology (*Zhang et al., 2020*). Key tumor-infiltrating immune cells, such as neutrophils, dendritic cells, T and B lymphocytes, and macrophages, directly or indirectly modulate tumor progression and influence clinical outcomes in various malignancies (*Domingues et al., 2016*; *Zhang & Zhang, 2020*). At present, GBPs are found to be implicated in the regulation host immune defense, cancer growth and metastasis (*Mustafa et al., 2018*; *Yu et al., 2020*). For instance, GBP2 exhibits anti-tumor effects on colorectal cancer and represents a potential immunotherapeutic target (*Yu et al., 2011*). As one of the IFN-γ-inducible genes, GBPs significantly upregulate the expression of IFN-γ in many cells, including T cells, NK cells, B cells (*Hunt, Kopacz & Vestal, 2022*). Simultaneously, these cells can infiltrate the tumor to be activated by IFN-γ. Lymph node-negative BRCA patients with a favorable prognosis have high levels of GBP1, 2, and 5 (*Hunt, Kopacz & Vestal, 2022*). Although it has been observed that GBPs play an important immunomodulatory role in a variety of tumors and is closely related to patients' prognosis, there is still a paucity of systematic studies probing into the expression patterns and functional mechanisms of GBPs in BRCA.

This study was the first to comprehensively assess the prognostic value of GBP family genes in BRCA and to construct a risk prediction model associated with GBPs. Based on the public databases, the expression characteristics of GBP family genes in BRCA and their relationship with prognosis were evaluated. Co-expression network was developed using WGCNA, and key genes related GBP scores were selected by differential analysis. The prognostic model was finally constructed by Cox and LASSO regression, and the immune infiltration characteristics and functional enrichment pathways of the key genes were further analyzed. Finally, the functional roles of the screened key genes were verified by cellular experiments. Overall, this study provides a new molecular basis for the prognostic assessment and immunotherapy treatment of BRCA.

## METHODS

### Data collection

We obtained BRCA clinical data and RNA-Seq data in FPKM from The Cancer Genome Atlas (TCGA) database (https://portal.gdc.cancer.gov/). This was achieved by removing samples without survival time or status, ensuring that the survival time of each patient was longer than 30 days but shorter than 10 years. Ultimately, 994 BRCA samples and 114 paraneoplastic control samples were included. Here, we focused on protein-coding genes due to their well-defined biological functions, stronger interpretability and comprehensive annotation information. Additionally, GSE20685 data was obtained from Gene Expression Omnibus (GEO, https://www.ncbi.nlm.nih.gov/geo/) database. In total, 327 BRCA samples were analyzed after the probes were converted to symbol using the annotation file while samples without overall survival (OS) data or clinical follow-up were eliminated. GSE20685 dataset and TCGA-BRCA dataset served as independent validation set and training set, respectively. Furthermore, seven GBPs were collected from previous research literature for further investigation (*Ning et al., 2023*).

### Differences in the mutation features of GBPs in BRCA samples

Based on the TCGA dataset, variations in the expressions of GBP genes between BRCA and paraneoplastic samples were analyzed. Next, ssGSEA (*Hanzelmann, Castelo & Guinney, 2013*) method in the R package "GSVA" was employed to calculate the GBP score for each sample in the TCGA_BRCA cohort, with the candidate GBPs as the background gene set. The BRCA samples were assigned into low- and high-GBP scores by the median value. The GBP gene mutations for each sample in the TCGA cohort were analyzed as genomic mutations are strongly linked to the progression of disease (*Zahn & Travis, 2015*). MuTect 2 software (*Beroukhim et al., 2010*) was utilized to process the mutation dataset of BRCA samples and paraneoplastic samples downloaded from the TCGA database and to map gene mutations in the low- and high-GBP subgroups.

### Construction of weighted gene co-expression network

Gene modules closely linked to the GBP score were classified by WGCNA. To more effectively detect strong correlations between modules, the pickSoftThreshold function of "WGCNA" was employed to decide the optimal soft threshold power ($\beta = 7$) (*Langfelder &*

*Horvath, 2008*). Under the threshold of minModuleSize = 60, hierarchical cluster analysis was performed to identify gene modules. According to the first principal components (PCs) of module expression, the "Heatmap" package (*Saunders, Liang & Li, 2007*), several module-trait genes were selected. Then, the correlation between module-trait genes and clinical trait for diagnosis was analyzed to evaluate the association between the modules and GBP scores. For modules showing the strongest module-trait relationships, the genes contained in the modules were tested for further analysis.

## Enrichment analysis

Subsequently, the "limma" package was employed to screen DEGs between tumor samples and paraneoplastic normal samples in the TCGA_BRCA cohort under $|\log_2 (\text{Fold Change})| > \log_2(2)$ with an adjusted $p < 0.05$ as the screening criteria. Then, genes ($p < 0.05$) in the intersection between the selected modular genes and DEGs were subjected to Gene Ontology (GO) and Kyoto Encyclopedia of Genes and Genomes (KEGG) functional analyses using the R package "clusterProfiler" (*Yu et al., 2012*; *Xia et al., 2024*). Top functions enriched in biological process (BP) term in GO analysis and the top enriched KEGG pathways were visualized into bar graphs.

## Risk modeling and validation

The intersecting genes were subjected to univariate Cox proportional risk regression in the R package "survival" (*Therneau & Lumley, 2015*) to select genes closely ($p < 0.05$) linked to the prognostic outcomes of BRCA patients in the TCGA_BRCA cohort (*Zhang et al., 2025*). The final genes for developing an effective risk model were refined applying 10-fold cross-validation and LASSO Cox regression analysis with the "glmnet" package (*Friedman, Hastie & Tibshirani, 2010*). The RiskScore model was developed by Multivariate stepwise regression analysis to screen important genes that showed a correlation coefficient independently linked to the prognosis of BRCA in the TCGA_BRCA dataset. The formula of the risk model was: RiskScore = $\Sigma\beta i \times Expi$, where $\beta i$ represents the multivariate regression Cox analysis coefficient of each gene, and $Expi$ represents the expression of each gene. Following zscore normalization, the optimal threshold value of Riskscore was used to divide the patients in the TCGA_BRCA dataset into low- and high-risk groups. The R package "survminer" (*Ozhan, Tombaz & Konu, 2021*) was then used to conduct survival analysis between high- and low-risk groups, and Kaplan–Meier (KM) survival curves were plotted for prognostic analysis. Furthermore, the diagnostic accuracy of the Riskscore in the TCGA-BRCA training set was assessed according to receiver operating characteristic analysis (ROC) using the R package "timeROC" (*Blanche, 2015*) and area under ROC curve (AUC) of 1-, 3- and 5-year. Next, the performance of the Riskscore was similarly validated in GSE20685.

## Functional characterization and immune infiltration analysis of BRCA patients in different risk groups

The DEGs between the two risk groups were filtered by the "limma" package under the filtering criterion $|\log_2 (\text{Fold Change})| > \log_2(2)$ and an adjusted $p < 0.05$. The DEGs were then subjected to functional annotation through GO and KEGG pathway

enrichment analyses using the DAVID online tool (https://davidbioinformatics.nih.gov/). Briefly, the DEGs were loaded into DAVID and the organism was selected as "Homo sapiens" to conduct GO (in three terms) and KEGG functional enrichment analysis. The gene-correlated pathways were examined by KEGG enrichment analysis. Only entries with $p < 0.05$ were selected.

The ssGSEA function of the R package "GSVA" was employed to calculate the scores of 28 types of immune cells (*Charoentong et al., 2017*) for the two risk subgroups, so as to analyze the correlation between Riskscore and immune function in different BRCA patients. The immune infiltration for the TCGA-BRCA dataset samples was assessed by calculating StromalScore, ImmuneScoreh and ESTIMATEScore with the R package "estimate" (*Yoshihara et al., 2013*). Then, we collected 29 gene signatures that represent the primary functional components of the tumor and immune, stromal, and other cell populations (*Bagaev et al., 2021*). The correlations between gene signatures and Riskscore were analyzed using ssGSEA (*Liu & Huang, 2023*).

## Cell culture and plasmid transfection

From the American Type Culture Collection (ATCC, Manassas, VA, USA), we ordered human BRCA cells (MDA-MB-231 and AU565) and human normal mammary epithelial cells (MCF-10A). DMEM medium (11965092, Gibco, Waltham, MA, USA) and RPMI 1640 (11875093, Gibco, USA) medium were used to culture MDA-MB-231 cells and AU565 cells, respectively. All the cultures were added with 1% penicillin streptomycin (15140122, Gibco, USA) and 10% heat-inactivated fetal bovine serum (FBS, 10099141, Gibco, USA). Cells were cultured at 37 °C and 5% $CO_2$ in an incubator.

Following the instructions, *DACT2* overexpression plasmid (oe-*DACT2*) or control plasmid (oe-NC) purchased from GenePharma (Shanghai, China) was transfected into MDA-MB-231 and AU565 cells in logarithmic growth phase ($2 \times 10^4$ cells/well) using Lipo3000 Liposome Transfection Reagent (L3000-001, Thermo Fisher Scientific, Waltham, MA, USA). After transfection for 48 h, the cells were collected for further study.

## RNA extraction and quantitative real-time PCR

Total RNA was separated from MDA-MB-231, MCF-10A, and AU565 cells using the RNA Extraction Kit (TRIzol, Invitrogen, Waltham, MA, USA), following the protocols. The purity and concentration of the total RNA was quantified, and the cDNA templates were generated using the HiScript II kit (R233-01, Vazyme, Nanjing, China). The qRT-PCR was conducted utilizing specific primers and the KAPA SYBR® FAST kit (Sigma Aldrich, St Louis, MO, USA). Data were processed using the $2^{-\Delta\Delta CT}$ method, with GAPDH as an internal control (*Zhang et al., 2023*). See Table 1 for the primer sequences used in the study.

## Cell viability

Following the instructions, CCK-8 assay (Dojindo, Rockville, MD, USA) was applied to assess the effect of *DACT2* on the viability of MDA-MB-231 and AU565 cells. Briefly, the cells were cultured for 0 (7,000 cells/well), 24 (5,000 cells/well), 48 (3,000 cells/well) and 72 (2,000 cells/well) h in 96-well microtiter plates. After washing the cells with PBS twice,

**Table 1  Primer sequences applied in this study.**

| Gene name | Forward primer | Revers primer |
|---|---|---|
| PSME2 | 5′CCTGGTTAAGCCAGAAGTCTGG 3′ | 5′CATTCACCCTCTCCAGCACCTT 3′ |
| DACT2 | 5′CTACACCAGGAGCGACTCAGAG 3′ | 5′ACTCACGGTCTCCGAATCGGTT 3′ |
| PIGR | 5′TACTGGTGTGGAGTGAAGCAGG 3′ | 5′AGCACCTTCTCATCAGGAGCAG 3′ |
| STX11 | 5′GAGATGAAGCAGCGCGACAACT 3′ | 5′CCAGCAAGTTCTCGGAAAACACG 3′ |
| GAPDH | 5′TTGCCCTCAACGACCACTTT 3′ | 5′TCCTCTTGTGCTCTTGCTGG 3′ |

100 µL of fresh culture media and 10 µL of CCK-8 solution were added into each well for cell incubation at 37 °C with 5% $CO_2$ for 3 h. The absorbance at 450 nm was read by SPECTROstar® Nano (BMG LABTECH GmbH, Ortenberg, Germany) (*Tang et al., 2024*).

## Cell migration and invasion assays

The impact of *DACT2* expression on the migration and invasion capacities of MDA-MB-231 and AU565 cells was also investigated. Wound healing assay was conducted to measure the cell migratory ability. Briefly, 6-well plates were added with transfected cells ($5 \times 10^5$/mL) and two mL of cell suspension for cell incubation at 37 °C with 5% $CO_2$. A 10 µL plastic pipette tip was utilized to scratch a uniform wound on the monolayer once the cell density reached approximately 80%. Following PBS wash, the monolayers were cultured in non-FBS medium. The wound edge between the two edges of the migrating cell sheet were measured and photographs at 0 and 48 h. Each experiment was carried out in triplicate. The upper transwell chamber coated with 10% Matrigel (Corning, Corning, NY, USA) contained $1 \times 10^5$ cells. After incubating the cells for 24 h, those still in the upper chamber were removed. After that, the cells in the lower chamber was fixed by 4% paraformaldehyde and colored by 0.1% crystal violet solution. Finally, the invaded or migrated cells were quantified from six distinct fields of view under a microscope.

## Statistical analysis

All the statistical analyses were conducted using R software version 3.6.0 (R Foundation, Vienna, Austria) and GraphPad Prism 8 (GraphPad Software, San Diego, CA, USA). Differences between continuous variables in the two groups were calculated by Wilcoxon rank-sum test. Correlations were computed by Spearman method, and the log-rank test was applied to compare survival differences between two groups of patients. In cellular assays, multiple group comparisons were performed using Student's *t*-test or two-way ANOVA. For data that did not conform to normal distribution, non-parametric tests such as Mann–Whitney U test or the Kruskal–Wallis test were used for statistical analysis. A $p < 0.05$ stood for statistically significant difference.

## RESULTS

### GBP gene expression and mutation in BRCA

In the TCGA cohort, analysis on the expressions of GBPs showed that the levels of GBP2, GBP4 and GBP6 were lower in BRCA samples than in paraneoplastic samples ($p < 0.0001$), while the expressions of GBP3 and GBP5 were notably higher in BRCA

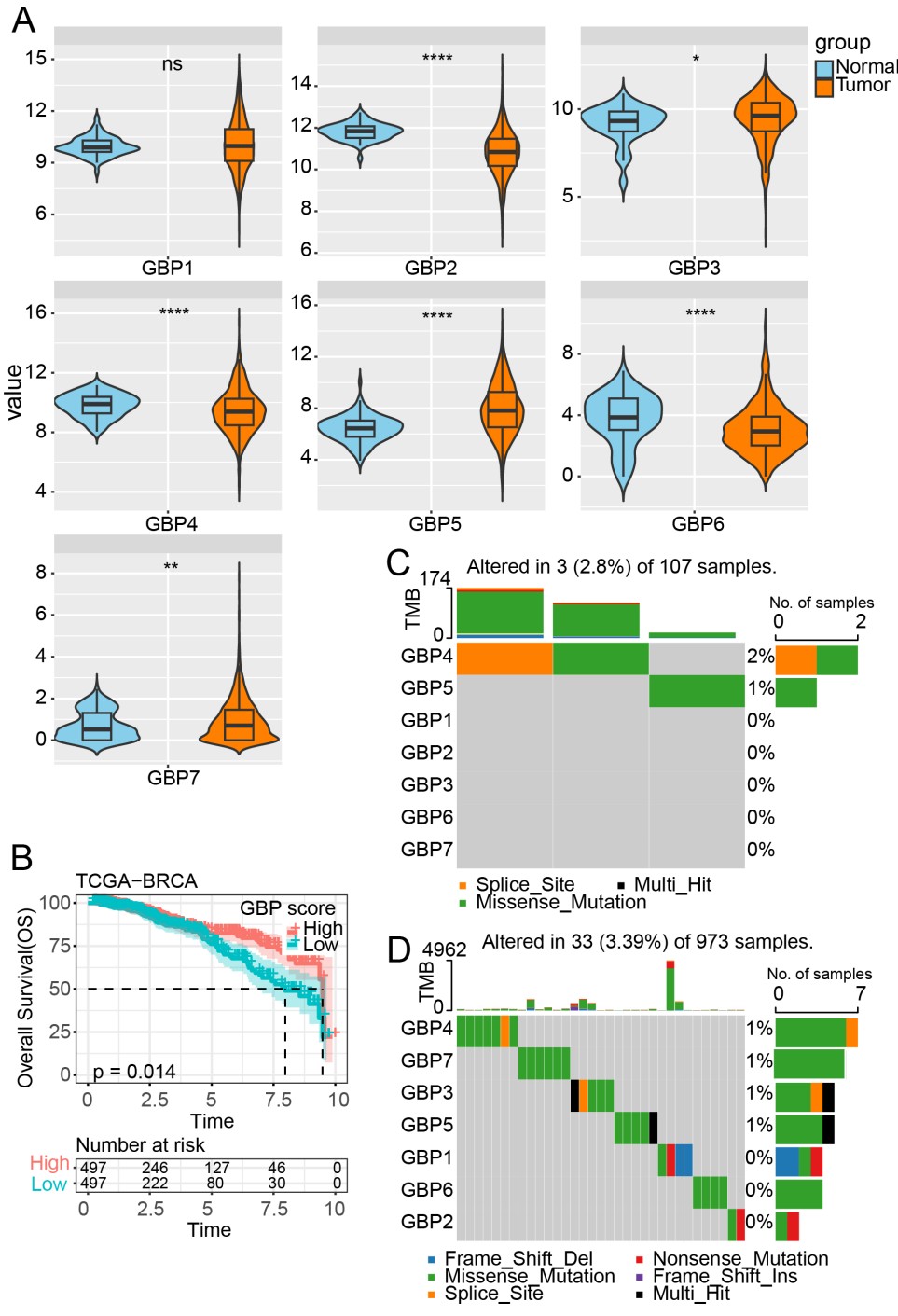

**Figure 1 Expression and mutation analysis of GBP gene in TCGA cohort.** (A) Differential analysis of GBP gene expression between BRCA samples and paraneoplastic samples in TCGA. (B) Relationship between GBP score and OS in BRCA patients. (C) Mutations of BRCA gene in paraneoplastic tissues. (D) Mutations of BRCA gene in BRCA tissues. ns indicates $p > 0.05$; $*p < 0.05$, $**p < 0.01$, and $****p < 0.0001$.

samples than paraneoplastic samples ($p < 0.05$, Fig. 1A). The GBP score for each sample in the TCGA_BRCA dataset was computed by the ssGSEA method, and the patients were divided by the median value into high and low GBP score groups. Prognostic study demonstrated that BRCA patients with high GBP scores had longer survival time, whereas those with low GBP values had a worse OS ($p = 0.014$, Fig. 1B). Next, analysis on the GBP gene mutations between BRCA and paraneoplastic samples in TCGA cohort showed that only GBP4 (2%) and GBP5 (1%) genes were mutant in paraneoplastic tissues (Fig. 1C). In contrast, BRCA tissues had numerous mutations in GBP3 and GBP5 genes in the same sample. Additionally, we identified mutations in GBP7 (1%) and GBP3 (1%) genes, along with the previously mentioned GBP4 (1%) and GBP5 (1%) (Fig. 1D).

## WGCNA identified GBP-related gene modules

Gene expression modules linked to GBPs in the TCGA-BRCA cohort were identified by the R package "WGCNA". The soft threshold power of 7 was selected to develop a topological network to ensure the scale-free topology of the network (Fig. 2A). Ultimately, we obtained 16 co-expression modules, with 60 genes in each module at least (Fig. 2B). Notably, the number of genes in the magenta, grey, and turquoise modules was comparatively high (Fig. 2C). Genes in the grey module could not be merged into other modules. In order to select clinically significant modules, we calculated the correlation of each module with GBP score and plotted a heat map of module-shape correlation. Light cyan and magenta modules were closely positively connected with the GBP score (cor $= 0.73$, $p = 1.42e-165$; cor $= 0.77$, $p = 3.97e-192$, Fig. 2D). Moreover, a strong association between MM and GS was detected in the lightcyan module (cor $= 0.72$, $p = 3.6e-14$) and magenta module (cor $= 0.9$, $p < 1e-200$) (Figs. 2E–2F).

## Differential gene analysis and functional characterization

Subsequently, we identified DEGs between BRCA samples and paraneoplastic samples in the TCGA dataset and finally screened 4,215 DEGs (Fig. 3A). Subsequently, we found 393 present in the intersection between the lightcyan and magenta module genes and the DEGs (Fig. 3B). GO and KEGG was performed on the common genes to investigate their regulatory functions in BRCA pathogenesis. GO-BP analysis showed significant enrichment of the intersecting genes in immune-related functions (Fig. 3C), including leukocyte migration, T-cell activation, positive regulation of cytokine production. The KEGG analysis revealed that the intersecting genes were considerably enriched in the viral protein interaction with cytokines and cytokine receptors and cytokine-cytokine receptor interaction pathways (Fig. 3D).

## Establishment of a risk model and verification

The intersecting genes were subjected to univariate Cox regression to identify GBP-linked DEGs with the most significant influence on the prognostic ourcomes of BRCA in TCGA-BRCA while removing redundant confounding genes. The screened genes were then further compressed based on LASSO analysis (Figs. 4A–4B). Multivariate stepwise regression analysis determined four distinctive genes (*PSME2*, *DACT2*, *PIGR* and *STX11*) independently linked to the prognosis of TCGA-BRCA patients (Fig. 4C). Then, the

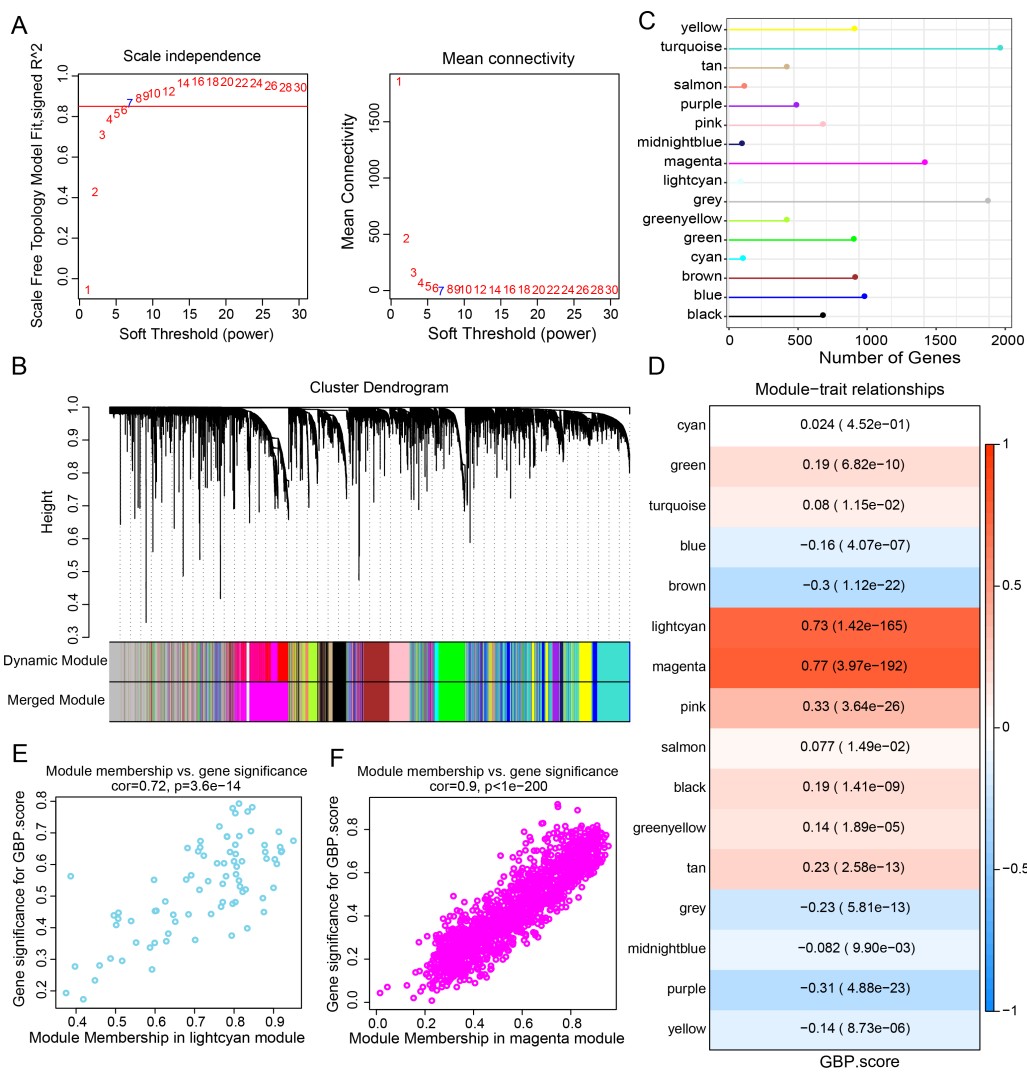

**Figure 2  Development of a co-expression network for the TCGA-BRCA cohort.** (A) Scale-free fit index analysis on different soft threshold powers (β), which were subjected to average connectivity analysis. (B) Gene dendrogram based on dissimilarity metric (1-TOM) clustering. (C) Gene number in each module. (D) Correlation of module eigenvectors with features for each module. (E) Gene significance *vs.* module membership for GBP-related genes score in the lightcyan module was visualized into a scatter diagram. (F) Gene significance *vs.* module membership for GBP-related genes score in the magenta module was visualized into a scatter diagram.

prognostic outcomes of TCGA-BRCA patients was predicted by the risk model: RiskScore $= (-0.28*PSME2) + (-0.157*DACT2) + (-0.066*PIGR) + (-0.237*STX11)$. Additionally, based on the optimal critical value of RiskScore, TCGA-BRCA patients were classified into low- and high-risk groups. The OS of low-risk TCGA-BRCA patients was more unfavorable than that the high-risk patients, as shown by the KM curves ($p < 0.0001$, Fig. 4D). Utilizing the "timeROC" R package, ROC analysis for 1-, 3-, and 5- year prognostic prediction was performed to evaluate the effectiveness of RiskScore in prognostic characterization. The AUC results verified the classification accuracy of the RiskScore, with an AUC value of

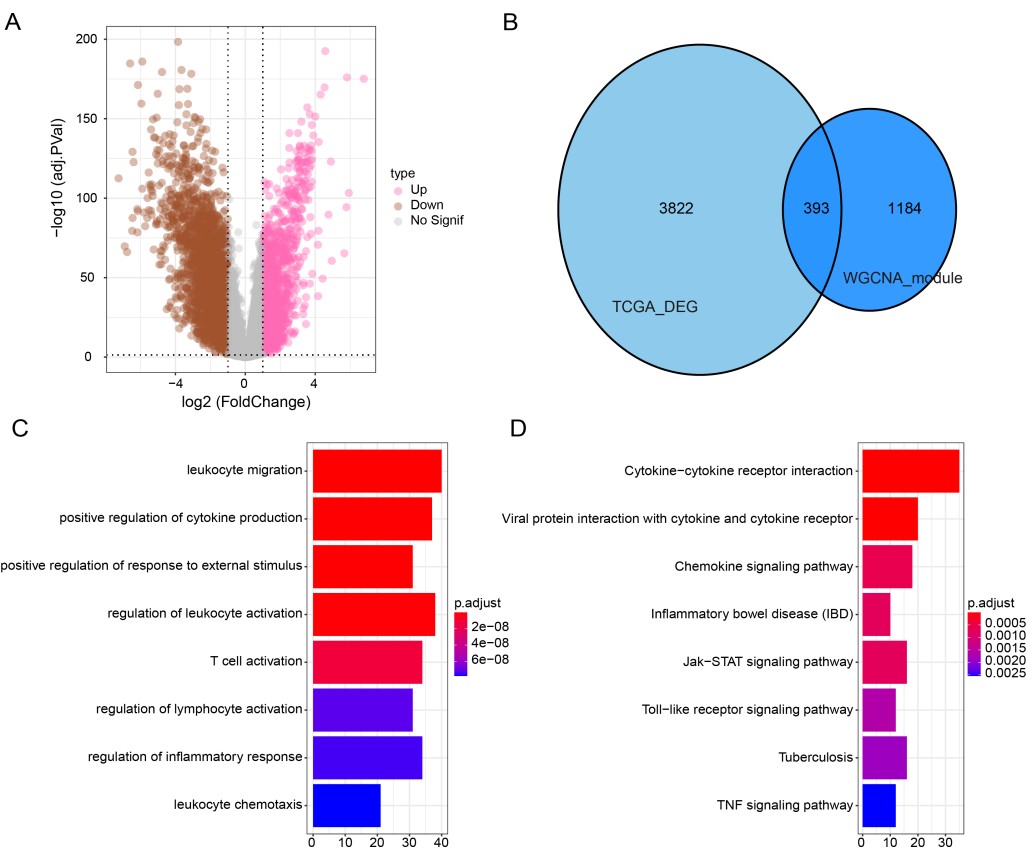

**Figure 3  Functional enrichment analysis of DEGs with modular gene intersection genes.** (A) Volcano plot of DEGs between BRCA samples and paraneoplastic samples in the TCGA-BRCA dataset. (B) The intersection of DEGs with lightcyan and magenta module genes. (C-D) GO and KEGG functional enrichment analysis of the intersected genes.

0.73, 0.66 and 0.69 for 1-, 3-, and 5- year prognostic prediction, respectively (Fig. 4E). Analysis on the expressions of the identified genes in TCGA-BRCA patients indicated that the high-risk BRCA group had lower expressions of *PSME2*, *DACT2*, *PIGR* and *STX11* than the low-risk group (Fig. 4F). We divided TCGA-BRCA patients by the median expression of the four genes into low and high expression groups, and examined the correlation between the expression of the four genes and patient prognosis. It was found that BRCA patients in the low expression groups of *PSME2* ($p = 0.00064$), *DACT2* ($p = 0.00015$), *PIGR* ($p = 0.00093$) and *STX11* ($p = 0.0001$) had poorer prognosis and shorter survival time (Fig. 4G).

The robustness of the RiskScore in the GSE20685 dataset was tested applying the RiskScore and equivalent coefficients used in the analysis on the training set. The validation set showed similar results to those in the training set that high-risk BRCA patients had a worse prognostic outcome than BRCA patients with a low risk ($p < 0.015$, Fig. 4H). The AUC value of 1-, 2-, 3-, 4-, and 5- year prognostic prediction in GSE20685 validation set reached 0.69, 0.6, 0.58, 0.58 and 0.57, respectively (Fig. 4I). All the four prognostic genes

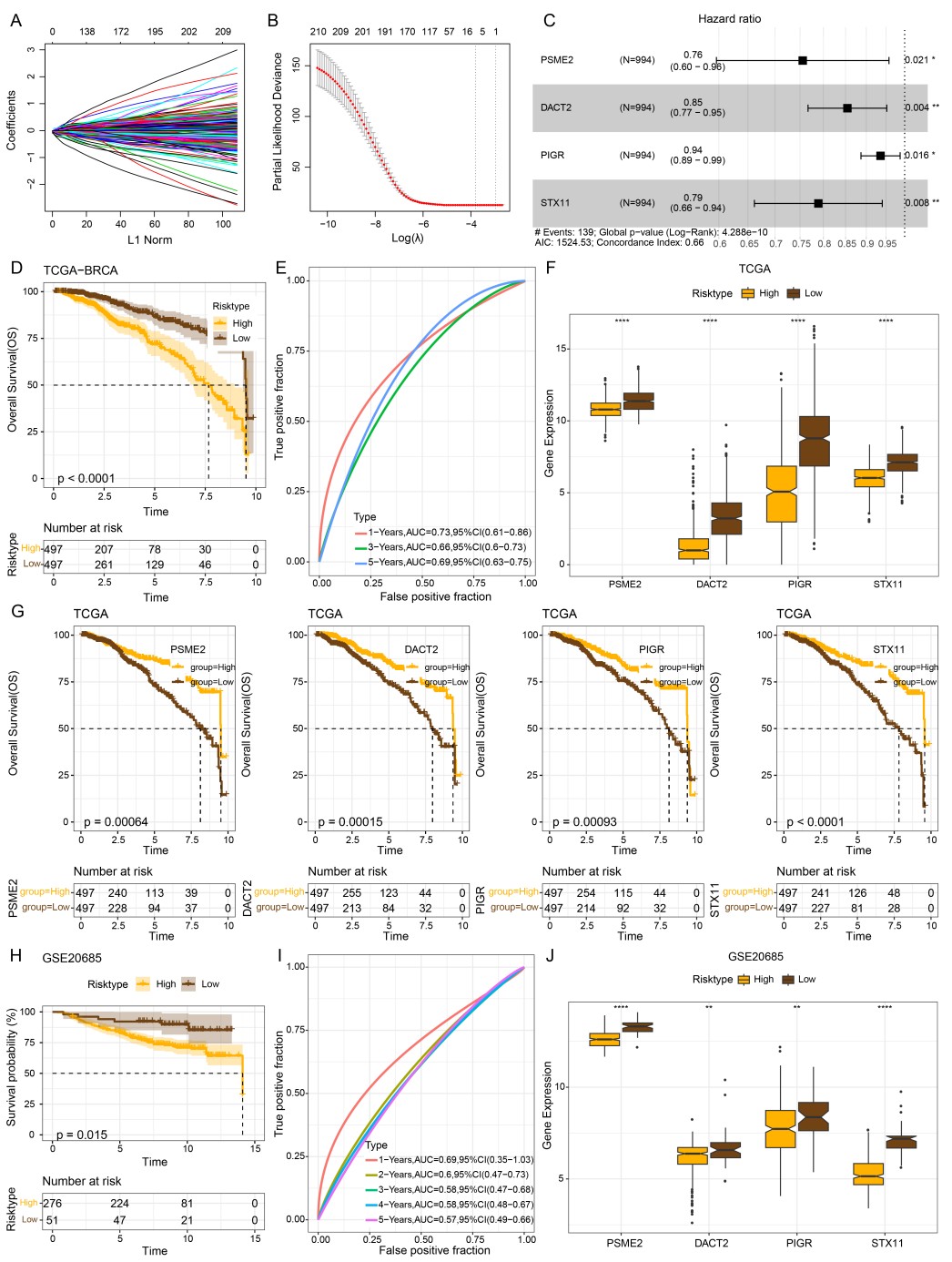

**Figure 4 Establishment of a prognosis model for BRCA patients and validation.** (A) Path diagram of LASSO coefficients for pivotal genes in TCGA-BRCA training with GBP. (B) Cross-validation curves in LASSO regression. The optimal λ value in the training group determined by 10-fold cross-validation. (C) Multivariate random forest plot. (D) The TCGA training data cohort was plotted with KM survival curves. (E) ROC curves for the performance of the RiskScore in the TCGA training data cohort. 

**Figure 4 (…continued)**
(F) Expressions of the prognosis genes in the TCGA cohort. (G) KM survival curves for the prognosis genes. (H) KM survival curves was plotted to reflect the performance of the RiskScore in the GSE20685 cohort. (I) ROC curve of RiskScore in the GSE20685 cohort. (J) The levels of prognosis genes in the GSE20685 cohort. $**p < 0.01$ and $****p < 0.0001$.

were low-expressed in the high-risk category in the GSE20685 validation cohort (Fig. 4J), suggesting the accuracy of these genes as prognostic predictors for BRCA.

## TCGA-BRCA enrichment analysis on the DEGs between the two risk groups

Next, 813 DEGs were filtered between low- and high-risk subgroups. The GO-BP enrichment analysis revealed that these DEGs were mainly implicated in inflammatory response, signal transduction, inflammatory response immune response, adaptive immune response, and adaptive immune response pathways (Fig. 5A). In CC terms, these genes mainly localized at the extracellular space, cell surface, external side of the plasma membrane, and some other structures (Fig. 5B). In MF terms, these genes were closely involved in signaling receptor activity and cytokine activity (Fig. 5C). The DEGs among different risk subgroups of BRCA mainly influenced cytokine activity, hematopoietic cell lineages, cell adhesion molecules, transmembrane signaling receptor activity, viral protein-cytokine, cytokine-cytokine receptor interactions, and cytokine receptor interactions (Fig. 5D).

## Relationship between RiskScore and immune characteristics

Immune cell infiltration in TCGA-BRCA patients was analyzed to examine the differences in the immunological milieu of the patients across risk groupings. Analysis of the infiltration abundance of 28 immune cells in BRCA patients revealed that high-risk BRCA patients had higher infiltration of activated CD8 T cells, activated CD4 T cells, activated B cells than the low-risk group, while the infiltration of mast cells, macrophages, and effector memory CD8 T cells was lower in the high-risk group (Fig. 6A), indicating that high-risk BRCA patients had lower immunoreactivity. The ESTIMATE showed a negative correlation between RiskScore and StromalScore, ImmuneScore, and ESTIMATEScore (Fig. 6B). This suggested that immune cell infiltration may be lowered in patients in the BRCA high-risk group. Next, applying the ssGSEA method, it was observed that the majority of the 29 gene signatures were closely linked to *PSME2*, *DACT2*, *PIGR* and *STX11*. Furthermore, the RiskScore had a negative correlation with anti-tumor immune infiltrate but was irrelevant to tumor proliferation (Fig. 6C).

## *In vitro* cell-based model to validate the expressions and potential biological functions of the key signature genes

According to the results of qPCR, MDA-MB-231 and AU565 cells had remarkably higher *PSME2* levels than in MCF-10A cells, while both AU565 and MDA-MB-231 cells had significantly lower levels of *DACT2* and *STX11*. However, only AU565 cells had notably downregulated expression of *PIGR* (Fig. 7A). Research showed that *DACT2* could cause cellular G1/S phase inhibition and suppress BRCA cell proliferation, demonstrating a

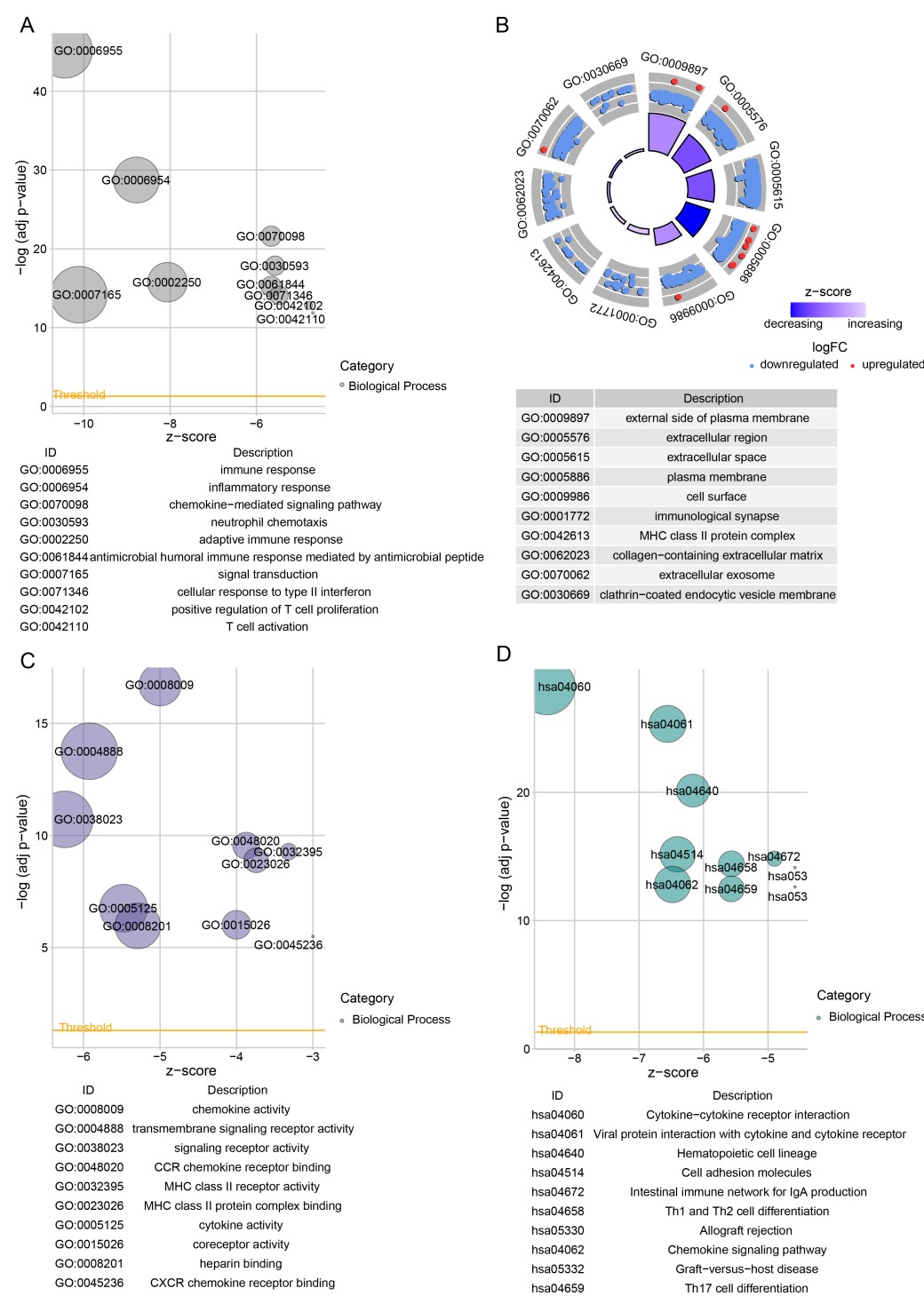

**Figure 5** **Differentially expressed genes between TCGA-BRCA low- and high-risk groups were subjected to functional enrichment analysis.** (A) Bubble plots for GO-BP enrichment analysis. (B) Bubble plots for GO-CC enrichment analysis. (C) Bubble plots for GO-MF enrichment analysis. (D) KEGG enrichment analysis bubble plots.

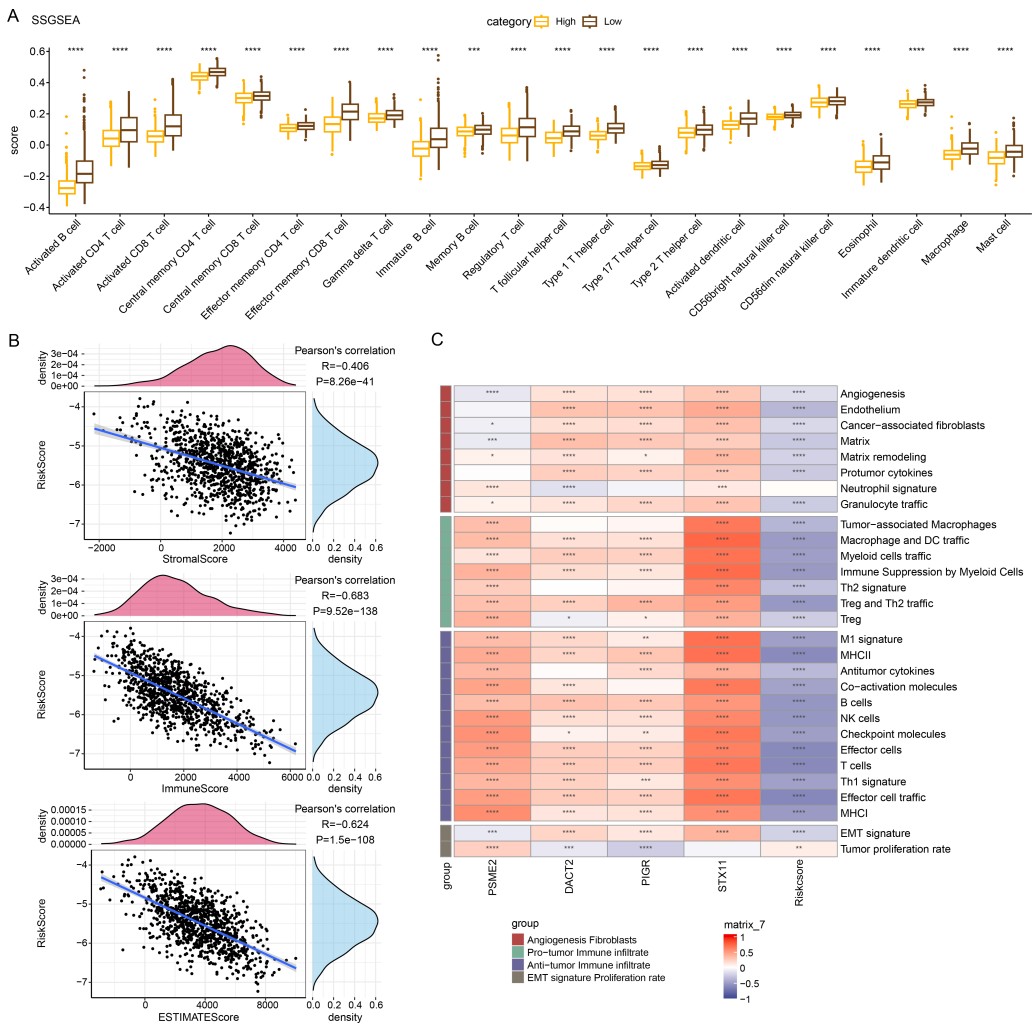

**Figure 6    Relationship between RiskScore and BRCA immunologic profile.** (A) ssGSEA assessment of immune infiltration in low-risk and high-risk groups. (B) Association between cellular infiltration score and RiskScore in ESTIMATE. (C) TCGA cohort RiskScore correlates with 29 gene signatures. ****$p <$ 0.0001, ***$p < 0.001$, **$p < 0.01$, *$p < 0.05$, ns indicates $p > 0.05$.

strong research basis and biological feasibility (*Li et al., 2017*). For this reason, we selected *DACT2* for further verification. The CCK-8 assay revealed that AU565 and MDA-MB-231 cell viability was considerably downregulated by *DACT2* overexpression (Figs. 7B–7C). The wound healing and transwell assays demonstrated that *DACT2* overexpression markedly suppressed MDA-MB-231 and AU565 cell migration and metastasis (Figs. 7D–7G).

## DISCUSSION

Highly heterogeneous nature of BRCA limits the reliability of using conventional clinicopathologic variables to predict patients' prognosis. Advancement of sequencing technology has allowed researchers to analyze the prognostic significance of molecular pathways in cancers and identify effective biomarkers (*Foote et al., 2023*). This study

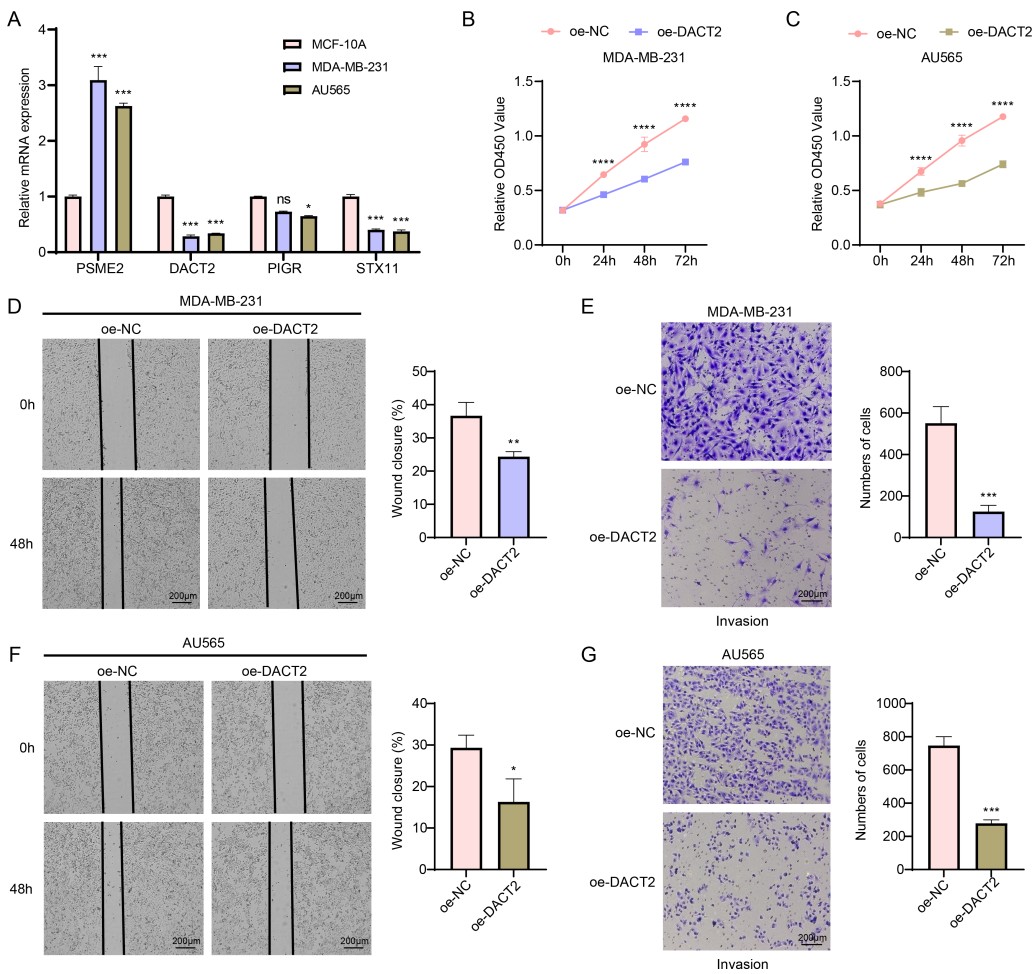

**Figure 7** **Exploring the biological role of *DACT2* in BRCA.** (A) The expressions of *PSME2*, *DACT2*, *PIGR* and *STX11* in MCF-10A, MDA-MB-231 and AU565 cells were detected by qPCR. (B–C) Verification of the effect of *DACT2* overexpression on the viability of MDA-MB-231 and AU565 cells. (D–G) Statistical analysis of representative images and invasive cell counts of AU565 and MDA-MB-231 cells after *DACT2* overexpression in wound healing assay and transwell assay. All procedures were conducted in three independent replicated experiments. Data were shown as SD ± mean, ****, $p < 0.0001$, ***, $p < 0.001$; **, $p < 0.01$; *, $p < 0.05$; ns, $p > 0.05$, not statistically significant.

measured the expressions of GBP genes and their relationship with the prognostic outcomes of BRCA based on public databases. The key modular genes linked to GBP score were identified by performing WGCNA and differential analysis, and a prognosis model was finally developed using *PSME2*, *DACT2*, *PIGR* and *STX11*. The results demonstrated that high GBP scores were predictive of longer survival, and that the low-risk group had higher immune cell infiltration. *In vitro* experiments revealed that *DACT2* significantly suppressed the activity, invasive and migratory capabilities of BRCA cells. This study confirmed the potential prognostic value of GBP-correlated genes in BRCA and their close relation with the immune microenvironment, providing new understanding for individualized treatment in BRCA.

We created a risk model to accurately assess the prognosis of BRCA using a machine learning approach to identify gene modules associated with GBPs. Based on the DEGs between BRCA tissues and para-cancerous tissues, *PSME2*, *DACT2*, *PIGR* and *STX11* were determined as the signature genes potentially related to the prognosis of BRCA. *PSME2* is primarily linked to the assembly of pre-replicative cell complexes (*Choi et al., 2019*; *Jayadev & Yusuff, 2024*). According to the prognostic characterization of BRCA prognosis-related gene constructs including *PSME2*, high-risk BRCA patients had higher levels of mutational landscapes, lower immune infiltration and an unfavorable prognosis. High-risk BRCA patients responded to chemotherapy but were resistant to immunotherapy (*Fang et al., 2023*). BRCA cells were unable to proliferate and create clones when *PSME2* is downregulated (*Qiu et al., 2024*). According to Huang et al., BRCA cells had low *DACT2* expression. Upregulating *DACT2* prevents glycolysis and increases mitochondrial oxidative phosphorylation in BRCA, which could inhibit the malignant transformation of BRCA cells (*Huang et al., 2021*). *DACT2* suppresses tumor formation in xenografted mouse BRCA cells by inhibiting BRCA cell growth and inducing G1/S-phase blockage in BRCA cells (*Li et al., 2017*). Though upregulated expression of polymeric immunoglobulin receptor (*PIGR*) may indicate the polarized state of tumor-associated immune cells in BRCA (*Asanprakit et al., 2022*), differential expression of PIGR as a transporter of polymeric immunoglobulins across epithelial cells in BRCA cells may not necessarily impact cellular and cellular behaviors (*Asanprakit et al., 2023*). *STX11* is abundant in immune cells, while TNFα release and phagocytosis of apoptotic cells and antibody-dependent target cells are promoted by silencing *STX11* in macrophages (*Zhang et al., 2008*). Compared to normal samples, *STX11* expression is lower in BRCA samples (*Dong, Li & Zhuang, 2024*). These findings indicated that GBPs might be viable gene targets for prognostic evaluation and the development of personalized BRCA therapies.

Inflammatory cytokines, which belong to the IFN γ-stimulated superfamily, also trigger GBPs in immune cells (*Lubeseder-Martellato et al., 2002*). Inflammatory cytokines may contribute to inflammatory activity by inducing GBPs. According to a retrospective study, patients with rheumatoid arthritis, psoriasis, inflammatory bowel disease, and some other conditions have elevated levels of GBPs in their serum and afflicted tissues (*Haque et al., 2021*). In hepatocellular carcinoma studies, tumor samples with high GBP scores also manifest significant inflammatory features, less tumor proliferation, and more immune-related features (*Ning et al., 2023*). By regulating this pathway, cytokine-cytokine receptor interactions implicated in adaptive and innate inflammatory host defense, cell death, and other processes may prevent the development of BRCA (*He et al., 2024*). Modification of bone marrow hematopoietic spectrum potential reduces bone metastases from BRCA, and recurrence-free survival could be lowered by the presence of disseminated tumor cells in bone marrow (*Ubellacker et al., 2018*). In tumor cells, some biological processes, including cell adhesion, are mostly uncontrollable and are linked to the development of invasion and metastasis (*Sathyanarayana et al., 2003*). Cell adhesion-related genes are considered as promising therapeutic targets for the prevention of human BRCA because they are particularly prevalent in advanced BRCA stages and are closely linked to the tumor growth (*Calaf et al., 2022*). We conducted GO and KEGG enrichment analyses to better examine

the biological significance of the GBP genes in BRCA. It was found that the most involved processes of DEGs between low- and high-risk BRCA groups were immune response and inflammatory response, which are mainly localized on structures such as cell membranes and extracellular fluids. These DEGs were chiefly related to the biological functions such as transmembrane signaling receptor activity and chemokine activity. These findings demonstrated that the interaction of these pathways with the four prognosis genes may be critically involved in a worse prognosis of the high-risk BRCA patients.

Tumor microenvironment is a complex and dynamic ecology that mainly consists of immune cells, tumor cells, and supportive cells, and is essential for the development, spread, and metastasis of cancers (*Arneth, 2019*). This study found that the immune scores in the low-risk BRCA group were higher than those of the high-risk group. These findings aligned with previous studies, which demonstrate that enhanced immune infiltration is correlated with favorable clinical outcomes (*Sui et al., 2020*). Furthermore, the low-risk group had high infiltration of M1 macrophages and CD8 T cells. According to earlier research, these immune cells also have strong anticancer and immuno-boosting properties (*Ali et al., 2014*; *Mehta et al., 2021*). CD4 T cells realize its antiproliferative effect through the advancement of the cancer cell cycle in G1/S. Furthermore, BRCA growth is inhibited *in vivo* by CD4 T cells alone, and T cell-dependent tumor regression is induced *via* indirect pro-inflammatory/immune impact (*Seung et al., 2022*). A transcriptome data analysis reported that B cell profiles in BRCA samples are linked to better survival (*Hu et al., 2021*). On the other hand, B lymphocytes have been shown to have a favorable association with high histological-grade cancers and can accelerate the growth of tumors (*Guan et al., 2016*), which might be explained by various states of B cells. The BRCA low-risk group in our study had higher infiltration of 28 types of immune cells, including activated CD4 T cells, activated CD8 T cells, and activated B cells. This indicated that the low-risk BRCA group may have higher immunoreactivity. StromalScore, ImmuneScore and ESTIMATEScore were all negatively correlated with RiskScore, indicating that BRCA patients in the high-risk group may have lower immune cell infiltration. These findings suggested that low immune infiltration in the high-risk group could contribute to poor clinical prognosis of BRCA patients.

This research had several limitations. Firstly, this study was mainly based on transcriptomic data from TCGA and GEO databases, and lacked validation on multi-center and multi-population clinical samples. In the future, prospective studies should be conducted to improve the generalization and clinical applicability of the model by combining clinical cohorts from multiple centers, different regions and different populations. Secondly, despite validation through external datasets, there was still a lack of experimental and follow-up data from clinical samples. Our further study will carry out prospective follow-up analysis using clinical samples and validate the correlation between the clinical outcomes and the expressions of model genes applying immunohistochemistry and other techniques. Finally, as our immune cell infiltration analysis relied on algorithmic analysis, we plan to collect clinical tissue samples and perform immunohistochemical staining or flow cytometry to further confirm the relationship between the RiskScore and immune cell infiltration.

## CONCLUSION

The significance of GBP family genes in BRCA was investigated based on their expression pattern. Four GBP-related signature genes (*PSME2*, *DACT2*, *PIGR* and *STX11*) were integrated into a RiskScore. Comprehensive validation analyses confirmed that the model had a strong performance in terms of the prognosis, biological function, and immunological infiltration in BRCA. The current findings may contribute to the individualized therapy for patients suffering from BRCA.

**Abbreviations**

| | |
|---|---|
| **BRCA** | breast cancer |
| **GBP** | Guanylate-binding protein |
| **TCGA** | The Cancer Genome Atlas |
| **GEO** | Gene Expression Omnibus |
| **DEGs** | differentially expressed genes |
| **WGCNA** | Weighted gene co-expression network analysis |
| **GO** | Gene ontology |
| **KEGG** | Kyoto Encyclopedia of Genes and Genomes |
| **BP** | Biological process |
| **MF** | Molecular function |
| **CC** | Cellular component |
| **AUC** | Area under ROC curve |
| **DCA** | Decision curve analysis |
| **LASSO** | Least absolute shrinkage and selection operator |
| **OS** | Overall survival |
| **ROC** | Receiver operating characteristic analysis |
| **GSVA** | Gene set variant analysis |
| **ssGSEA** | Single-sample gene set enrichment analysis |

### Funding

The study was supported by National Natural Science Foundation of China (No. 81672754) and Key Science and Technology Projects in the Healthcare System of Nanshan District (No. NSZD2024071). The funders had no role in study design, data collection and analysis, decision to publish, or preparation of the manuscript.

### Grant Disclosures

The following grant information was disclosed by the authors:
National Natural Science Foundation of China: No. 81672754.
Key Science and Technology Projects in the Healthcare System of Nanshan District: No. NSZD2024071.

### Competing Interests

The authors declare there are no competing interests.

## Author Contributions

- Min Wei conceived and designed the experiments, performed the experiments, analyzed the data, prepared figures and/or tables, authored or reviewed drafts of the article, and approved the final draft.
- Peng Sun performed the experiments, authored or reviewed drafts of the article, and approved the final draft.
- Xuemei Liu conceived and designed the experiments, analyzed the data, prepared figures and/or tables, and approved the final draft.
- Xuhua Liu performed the experiments, authored or reviewed drafts of the article, and approved the final draft.
- Jie Lei conceived and designed the experiments, performed the experiments, analyzed the data, prepared figures and/or tables, and approved the final draft.

## Data Availability

The datasets generated and/or analyzed during the current study are available at GEO: GSE20685.

The raw data is available in GitHub and Zenodo:

- https://github.com/MinWei75/Raw-data.git
- MinWei75. (2025). MinWei75/Raw-data: Raw data (v.1.1.0). Zenodo. https://doi.org/10.5281/zenodo.15074454.

## Supplemental Information

Supplemental information for this article can be found online at http://dx.doi.org/10.7717/peerj.20058#supplemental-information.

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
