# Peer review of "Expression characterization of the guanylate-binding protein gene family in breast cancer and its association with the immune microenvironment"

_PeerJ, doi:10.7717/peerj.20058_

## Round 0.1 · original submission · Major Revisions

Based on the comments from two reviewers, I recommend major revisions before the manuscript can be considered for publication. While Reviewer 1 suggested minor revisions, the substantial concerns raised by Reviewer 2 need to be thoroughly addressed. Please provide a point-by-point response to all reviewers' comments and revise your manuscript accordingly.

Reviewer 1 ·

Basic reporting

no comment

Experimental design

no comment

Validity of the findings

no comment

Additional comments

The study explores the role of Guanylate-binding protein (GBP) gene scores in breast cancer (BRCA), which is a relatively novel angle, particularly in relation to prognosis and immunotherapy. While GBP has been implicated in immune regulation, its specific role in BRCA remains underexplored. This work contributes to the growing interest in immune-related gene signatures for predicting clinical outcomes in BRCA patients. However, there are still some deficiencies in details in the manuscript.
1. Line 19-20, The research purpose of this article should be clearly stated. Line 25, “modular genes” should be “module genes”. Line 26, “multifactorial stepwise regression analyses” should be “multivariate stepwise regression analyses”. LASSO Cox regression analysis is also used in this article.
2. Line 36-37, What is the specific pathway of difference. Line 48-53, What are the mortality rate and recurrence rate of the patients after treatment.
3. While the background discusses the general roles of GBP proteins in immunity and cancer, it lacks a focused review of existing studies specifically related to BRCA. The statement “Rarely, though, has the function of GBP in BRCA been documented” could benefit from citing relevant prior work (even if limited) to support this claim and highlight the novelty of the current study. Line 63-64, Do only GBP1 and GBP2 play a role in immunity.
4. Line 82-83, “Our goal in this work was to thoroughly examine how seven GBP family compounds affect BRCA.” This seems not to fully indicate the purpose of this article; these several genes are merely potential candidates as molecular markers.
5. The background mentions that TNM staging and molecular subtypes are currently the main determinants of prognosis in BRCA, but there is little discussion on the limitations of these systems, such as poor predictive accuracy in certain subgroups or inability to reflect tumor heterogeneity. A more detailed critique would better justify the need for a novel GBP-based prognostic model. Line 84-95, The end needs to present some important results to highlight the research significance of this paper.
6. Line 100, “samples with survival time and status were removed”, which appears contradictory. If the study involves survival analysis, excluding samples with survival data would compromise the validity of the prognostic model. Please clarify this statement. Line 101-102, The sentence “only the genes encoding the proteins were kept” lacks sufficient explanation.” Why were non-coding RNAs or other gene types excluded? What was the biological or methodological rationale for focusing only on protein-coding genes?
7. Line 130-140, The manuscript lacks a clear link between the functional enrichment and the overall study objective (i.e., understanding the role of GBP-related genes in BRCA prognosis and immunity).
8. Lien 158, RiskScore = ∑(βi × Expi), where βi is the regression coefficient of gene i, and Expi is the expression level of gene i. A more formal mathematical notation and explanation of normalization steps would improve the scientific rigor and readability of the model description. Line 277-278, Detailed methods should not appear in the results.
9. Line 355-369, Briefly summarize the background and purpose, and the major findings of this study should be well-described.
10. What is the specific clinical application of RiskScore and how does it improve patient outcomes. Line 455-440, This article has many limitations, but the author has not listed them all. This article needs polishing to further enhance its readability.

Reviewer 2 ·

Basic reporting

no comment

Experimental design

no comment

Validity of the findings

1. In Fig. 1, the labels “GBP1,” “GBP2,” etc., appear twice, which is confusing. Please remove the duplicates.
2. The legend of Fig. 1 contains *** symbols, but no significant symbols are found in the figure itself. Please remove them.
3. Fig. 4 has the same issue. Please specify in the legend the meanings of the significant symbols that appear in the figure. Do not include those that do not appear.
4. Why are only follow-up experiments conducted on DACT2 in Fig. 7? What about the other genes? Why not verify the functions of the other genes?
5. The legend should indicate the sample size, such as n=6.
6. What statistical methods were used for the results of cell proliferation and migration? The Statistical tests section does not describe the statistical methods used for data that follow a normal distribution. Please supplement this information.
7. The references are outdated. Please use literature from the past three years.
8. What do “2003” and “USA” mean in “From the American Type Culture Collection (ATCC, Manassas, Virginia, USA). 2003, ATCC, USA)?”
9. The instruments and reagents should include catalog numbers, such as “HiScript II kit (Vazyme, China).”

---

## Round 0.2 · accepted · Accept

Thank you for your thorough revisions. Both reviewers have recommended acceptance, and I am pleased to inform you that your manuscript has been accepted for publication.

Reviewer 1 ·

Basic reporting

The study explores the role of Guanylate-binding protein (GBP) gene scores in breast cancer (BRCA), which is a relatively novel angle, particularly in relation to prognosis and immunotherapy. While GBP has been implicated in immune regulation, its specific role in BRCA remains underexplored. This work contributes to the growing interest in immune-related gene signatures for predicting clinical outcomes in BRCA patients.

Experimental design

no comment

Validity of the findings

no comment

Reviewer 2 ·

Basic reporting

no comment

Experimental design

no comment

Validity of the findings

no comment

Additional comments

This study is the first to systematically explore the prognostic value of the guanine nucleotide-binding protein (GBP) gene family in breast cancer (BRCA), and has constructed a risk prediction model based on GBP-related genes. This fills the gap in the systematic research of the GBP gene in breast cancer. The risk model and immune association provide new tools for prognosis assessment. The results are highly original, especially in integrating bioinformatics and experimental verification, which is superior to existing literature. The manuscript structure is clear, the language is professional, and the charts effectively support the conclusions. I recommend publication.